# Observation of stacking engineered magnetic phase transitions within moiré supercells of twisted van der Waals magnets

Senlei Li[1,9], Zeliang Sun[2,9], Nathan J. McLaughlin[3], Afsana Sharmin[4], Nishkarsh Agarwal [5], Mengqi Huang[1], Suk Hyun Sung [5], Hanyi Lu[3], Shaohua Yan[6,7], Hechang Lei [6,7], Robert Hovden [5], Hailong Wang [1], Hua Chen [4,8], Liuyan Zhao [2]✉ & Chunhui Rita Du [1,3]✉

Recent demonstrations of moiré magnetism, featuring exotic phases with noncollinear spin order in the twisted van der Waals (vdW) magnet chromium triiodide $CrI_3$, have highlighted the potential of twist engineering of magnetic (vdW) materials. However, the local magnetic interactions, spin dynamics, and magnetic phase transitions within and across individual moiré supercells remain elusive. Taking advantage of a scanning single-spin magnetometry platform, here we report observation of two distinct magnetic phase transitions with separate critical temperatures within a moiré supercell of small-angle twisted double trilayer $CrI_3$. By measuring temperature-dependent spin fluctuations at the coexisting ferromagnetic and antiferromagnetic regions in twisted $CrI_3$, we explicitly show that the Curie temperature of the ferromagnetic state is higher than the Néel temperature of the antiferromagnetic one by ~10 K. Our mean-field calculations attribute such a spatial and thermodynamic phase separation to the stacking order modulated interlayer exchange coupling at the twisted interface of moiré superlattices.

Recently, twisted two-dimensional (2D) van der Waals (vdW) magnets have emerged as a new member in the suite of moiré quantum materials, where the spin degree of freedom is controlled through the local stacking order of moiré superlattices[1–11]. As opposed to moiré quantum electronic matter such as twisted graphene[12–15] and transition metal dichalcogenides[14–16] that feature enhanced electronic interactions from moiré flat bands, moiré magnets rely on twist-engineered magnetic interaction competitions to realize unconventional magnetic orders and spin excitations over moiré wavelengths, for example, in twisted vdW magnet chromium triiodide $CrI_3$[1–5,7–9,11].

Few-layer $CrI_3$ shows stacking-dependent interlayer magnetic exchange coupling that is ferromagnetic (FM) for the rhombohedral stacking and antiferromagnetic (AFM) for the monoclinic stacking geometry[8,17–19], whereas the intralayer exchange coupling is FM with a strong out-of-plane easy-axis anisotropy[8,17,18]. In twisted $CrI_3$ with spatially distributed rhombohedral and monoclinic stacking geometries, the competition between uniform FM intralayer exchange coupling and FM-AFM modulated moiré interlayer exchange interaction drives the formation of a range of novel magnetic orders, such as co-existing FM and AFM states within moiré supercells in twisted bilayer and

[1]School of Physics, Georgia Institute of Technology, Atlanta, GA 30332, USA. [2]Department of Physics, the University of Michigan, Ann Arbor, MI 48109, USA. [3]Department of Physics, University of California, San Diego, La Jolla, San Diego, CA 92093, USA. [4]Department of Physics, Colorado State University, Fort Collins, CO 80523, USA. [5]Department of Materials Science and Engineering, University of Michigan, Ann Arbor, MI 48109, USA. [6]Department of Physics, Beijing Key Laboratory of Optoelectronic Functional Materials MicroNano Devices, Renmin University of China, Beijing 100872, China. [7]Key Laboratory of Quantum State Construction and Manipulation (Ministry of Education), Renmin University of China, Beijing 100872, China. [8]School of Advanced Materials Discovery, Colorado State University, Fort Collins, CO 80523, USA. [9]These authors contributed equally: Senlei Li, Zeliang Sun. ✉e-mail: lyzhao@umich.edu; cdu71@gatech.edu

double trilayer CrI$_3$[1,2], and emergent magnetization and noncollinear spins in twisted double bilayer CrI$_3$[3–5]. So far, the ongoing research on twisted CrI$_3$ has mainly focused on investigating spatially modulated magnetic orders of the ground states[1–5,11], which represents only a subset of information about moiré magnetism. The local magnetic interactions, spin dynamics, and magnetic phase transitions within and across moiré supercells await exploration and are necessary for developing a comprehensive picture of moiré magnetism.

Here, we report scanning single-spin quantum sensing[20–22] of both static magnetization and dynamic spin fluctuations of moiré magnetism hosted by twisted double trilayer (tDT) CrI$_3$ across the second-order magnetic phase transition points ($T_c$). We show that the FM region within individual moiré supercells formed in small-twist-angle tDT CrI$_3$ exhibit a higher $T_c$ up to ~58 K in comparison with that of ~48 K for their AFM counterparts resulting in a nanoscale co-existing paramagnetic(PM)-ferromagnetic (FM) phase in an intermediate temperature regime (48 K < $T$ < 58 K), while such a phenomenon is absent in the large-twist-angle regime. Our experimental results are well explained by a proposed mean-field theoretical model of layer-resolved magnetic phases of tDT CrI$_3$ taking account of stacking engineered exchange interactions at the twisted interface. The current work highlights twist engineering as a promising tuning knob to realize local control of magnetic responses at individual stacking sites, which could contribute to a broad range of emerging 2D electronic applications[23,24]. The new insights on moiré magnetism presented in this study further highlight the potential of quantum metrology tools[20] in exploring unconventional spin-related phenomena in correlated magnetic quantum states of matter.

## Results

We first briefly review the pertinent material properties of tDT CrI$_3$, which provides a reliably high-quality moiré magnet platform for the current study[2]. Figure 1a shows a moiré superlattice structure formed by stacking two CrI$_3$ trilayers with a small twist angle. The local atomic registry exhibits a periodic modulation in real space, leading to spatially alternating stacking geometries on a length scale of moiré wavelengths[1–3]. At the monoclinic (AB') stacking site, the two CrI$_3$ trilayers are coupled by a positive exchange interaction $J_M$, leading to local AFM order at the twisted interface with a fully compensated net magnetic moment in the ground state[2,8]. In contrast, FM order with a net magnetic moment is established at the rhombohedral (AB) stacking site driven by a negative interlayer exchange interaction $J_R$[2,8]. Using first-principles calculations, a theoretical study has reported that the magnitude of $J_R$ could be one order of magnitude larger than that of $J_M$ due to the orbital-dependent exchange coupling[8]. Fundamentally, the sign and magnitude of magnetic interactions in twisted CrI$_3$ not only determine its local magnetic phases but also affect their dynamic responses to external perturbations. The former has been predicted and experimentally demonstrated recently[2,8], and the latter is the focus of the current work.

We fabricated tDT CrI$_3$ devices by the standard "tear-and-stack" technique and encapsulated them with hexagonal boron nitride (hBN) nanoflakes[1–3,12,25]. The samples for transmission electron microscopy and quantum sensing measurements were fabricated separately because they required different substrates for individual measurement purposes (See Methods Section for details). Selected area electron diffraction (SAED) patterns (Fig. 1b) of a surveyed sample area of a tDT CrI$_3$ device shows sets of the fifth-order Bragg peaks with sixfold rotation symmetry. The local mean twist angle is measured to be 0.8° ± 0.1° by fitting 2D Gaussians to the diffraction peaks, giving a moiré period of ~50 nm. Such an intermediate twist angle ensures a decent, regular moiré lattice structure formed in tDT CrI$_3$ that is visible in transmission electron microscopy measurements. Figure 1c presents a bright-field transmission electron microscopy (BF-TEM) image, which shows the characteristic hexagonal superlattice structures with a

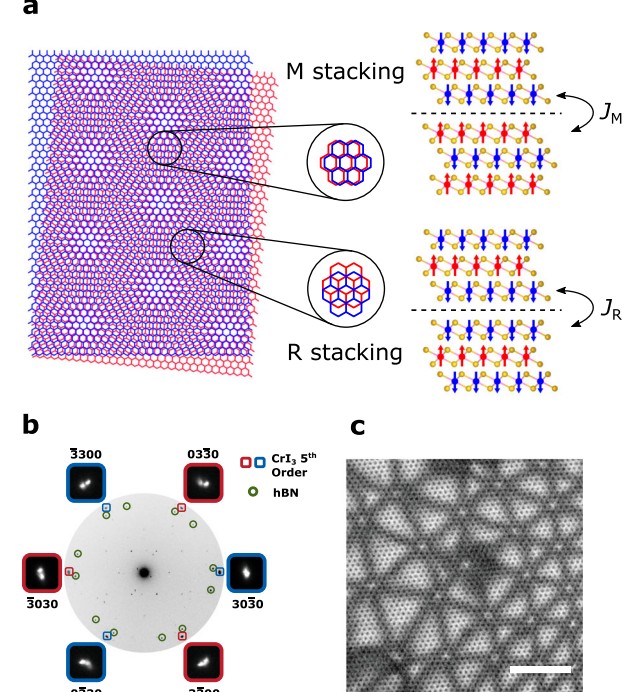

**Fig. 1 | Moiré superlattices of tDT CrI$_3$. a** Left: Moiré superlattice structure of a small-twist-angle tDT CrI$_3$. Only the two layers of Cr atoms adjacent to the twisted interface are shown for visual clarity. Cr atoms belonging to the top and bottom CrI$_3$ layers are labeled in blue and red colors, respectively. Right: Schematic of rhombohedral (AB) and monoclinic (AB') stacking driven FM and AFM orders in the magnetic ground state of small-twist-angle tDT CrI$_3$. Magnetic moments carried by the middle two CrI$_3$ monolayers are ferromagnetically or antiferromagnetically coupled depending on the local interlayer exchange interaction $J_R$ and $J_M$ at the twisted interface. The blue and red arrows represent the local magnetic moment carried by Cr atoms (blue and red balls) at individual layers. The yellow balls represent the I atoms and the black dashed lines highlight the twisted interface. **b** SAED patterns of the fifth-order Bragg peaks of a ~0.8° tDT CrI$_3$ device (blue and red rectangles) from a surveyed sample area of ~850 nm × ~850 nm. **c** BF-TEM real-space image of a sample region showing the characteristic hexagonal superlattice structure in the ~0.8° tDT CrI$_3$ device. The scale bar is 25 nm.

periodicity commensurate with the moiré wavelength. The distortions from the expected moiré lattice patterns could be induced by lattice strain, relaxation, and local structural inhomogeneities (see Supplementary Information Note 1 for details)[26].

Next, we utilize scanning nitrogen-vacancy (NV) microscopy[2,18,27–29] to spatially resolve the moiré magnetism hosted by tDT CrI$_3$ as illustrated in Fig. 2a. Scanning NV magnetometry exploits the Zeeman effect to quantitatively detect local magnetic stray fields longitudinal to the NV spin axis[21]. The magnitude of the magnetic field is directly related to the splitting of NV spin energies, which can be readout by optically detected magnetic resonance measurements[2,18] (see Supplementary Information Note 2 for details). The spatial resolution of scanning NV magnetometry is primarily determined by the NV-to-sample distance[30], which is ~70 nm in our measurements (see Supplementary Information Note 3 for details). In the current study, we report scanning NV quantum sensing measurements of a total of three twisted vdW magnet samples: 0.15° tDT CrI$_3$, 0.25° tDT CrI$_3$, and 15° tDT CrI$_3$. For the brevity of our narrative, 0.15° and 0.25° refer to the small-twist angle, and 15° refers to the large-twist angle in our description.

Figure 2b shows an optical microscope image of a prepared tDT CrI$_3$ sample with a calibrated local twist angle $\alpha = 0.15°$ and an expected moiré period of ~250 nm. Note that in order to visualize nanoscale magnetic patterns within individual moiré supercells, here $\alpha$ is chosen to be smaller than ~0.5°, ensuring that the resulting moiré

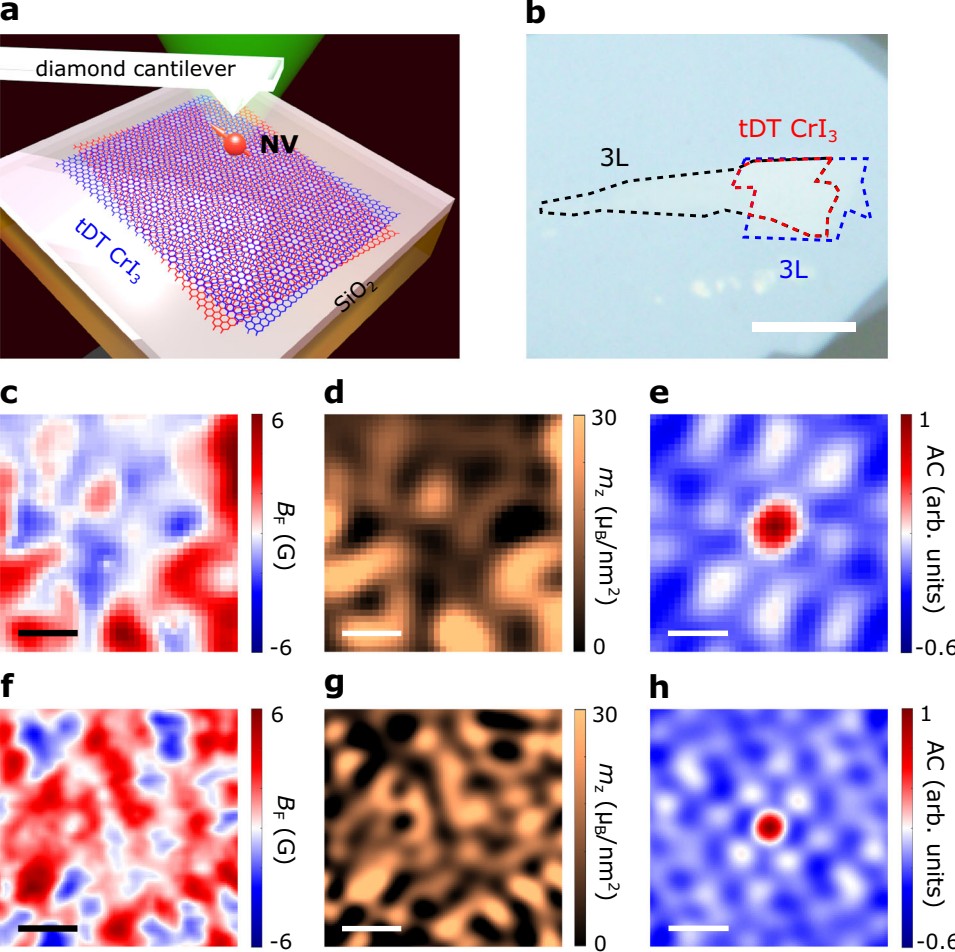

**Fig. 2 | Scanning single-spin magnetometry measurements of small-twist-angle tDT CrI₃. a** Schematic illustration of scanning NV measurements of twisted CrI₃. **b** Optical microscope image of a small-twist-angle tDT CrI₃ sample. The two torn trilayer CrI₃ flakes are outlined by the black and blue dashed lines, respectively, and the twisted area is highlighted by the red dashed lines. Scale bar is 10 μm. **c**, **f** Nanoscale scanning NV imaging of magnetic stray fields emanating from selected sample areas of a 0.15° tDT CrI₃ (**c**) and a 0.25° tDT CrI₃ device (**f**). **d**, **g** Magnetization maps reconstructed from the stray field patterns shown in **c** and **f** for the 0.15° tDT CrI₃ (**d**) and 0.25° tDT CrI₃ (**g**) sample. **e**, **h** Normalized auto-correlation (AC) maps of the stray field patterns shown in **c** and **f** for the 0.15° tDT CrI₃ (**e**) and 0.25° tDT CrI₃ device (**h**). Scale bar is 200 nm for images presented from **c** to **h**.

period is larger than our NV spatial sensitivity (~70 nm). Figure 2c presents a stray field $B_F$ map measured on a selected sample area of the 0.15° tDT CrI₃ device at 2 K. An external magnetic field of ~2000 G is applied along the NV spin axis in this measurement. The tDT CrI₃ sample shows clear multidomain features with stray fields of opposite polarity emanating from individual domains. The reconstructed out-of-plane magnetization $(m_z)$[2,18] map (Fig. 2d) manifests alternating FM and AFM patches on a length scale comparable with the moiré period (see Supplementary Information Note 4 for details). The local net magnetization is measured to be 0 and ~30 $\mu_B/nm^2$ for the AFM and FM domains, respectively, exhibiting the key feature of stacking-induced co-existing magnetic phases of moiré magnetism[1,2] (see Supplementary Information Note 5 for details). The measured local magnetization of FM order in 0.15° tDT CrI₃ also agrees with the theoretical value 29.4 $\mu_B/nm^2$[2,18]. By performing an autocorrelation operation[2] on the magnetization map, the periodic hexagonal shaped magnetic patterns of tDT CrI₃ is revealed in Fig. 2e, from which the local mean twist angle ($\alpha = 0.15°$) and moiré period (~250 nm) are confirmed. The notable alternating FM-AFM state is also observed in another 0.25° tDT CrI₃ device (Fig. 2f–h). It is worth mentioning that a larger twist angle naturally results in spatially more compact magnetic moiré patterns (Fig. 2g) with a reduced moiré period of ~150 nm (Fig. 2h).

We now present systematic scanning NV magnetometry measurements to show distinct magnetic phase transition temperatures for the observed FM and AFM states within individual moiré supercells. Figure 3a presents a zoomed-in magnetization map of a selected sample area of the 0.15° tDT CrI₃ device, showing two neighboring nanoscale FM and AFM domains. Due to the fully compensated net magnetic moment, temperature driven second-order magnetic phase transition of the AFM domain in tDT CrI₃ is challenging to access by measuring its emanating magnetic flux. To circumvent this issue, here we employ NV relaxometry method[27,31–35] to probe the intrinsic spin fluctuations at the local AFM (FM) regions. Figure 3b illustrates the mechanism of scanning NV relaxometry measurements, which takes advantage of the dipole-dipole interaction between local spin fluctuations of AFM (FM) spin density and a proximal NV center contained in a diamond cantilever. Spin fluctuations in a magnetically correlated system are driven by its time-dependent spin density distribution, for example, due to the dynamic imbalance in the thermal occupation of magnon bands with opposite chiralities[31,34,36]. For both AFM and FM systems with (un)compensated net static magnetic moment, spin-spin correlation-induced time-dependent fluctuations of the average spin density do not vanish and are expected to reach a maximum intensity around the magnetic phase transition points[33,34]. The emanating fluctuating magnetic fields at the NV electron spin resonance (ESR)

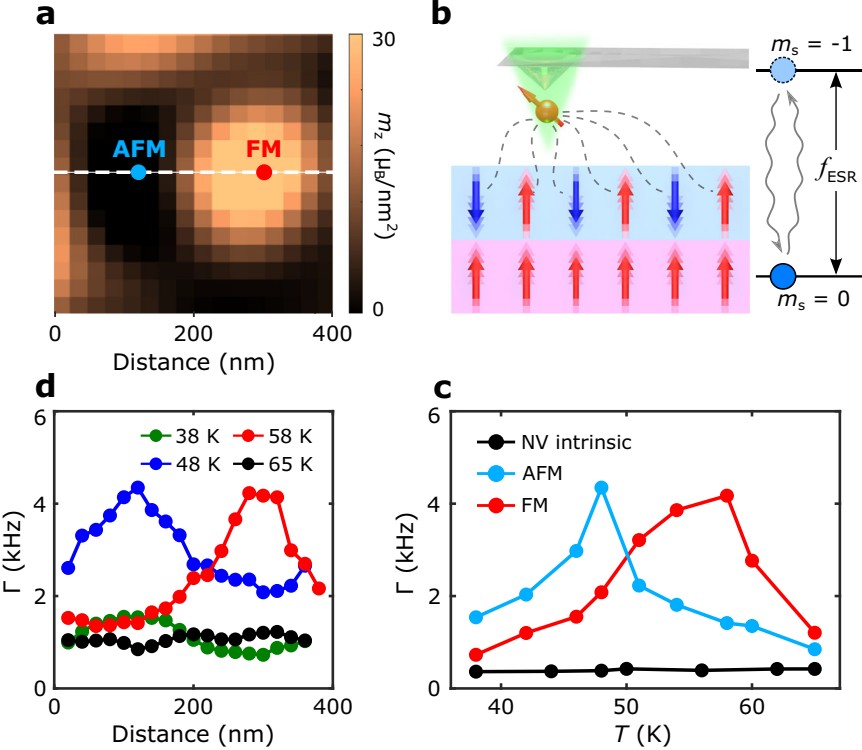

**Fig. 3 | NV spin relaxometry measurements of spin fluctuations in 0.15° tDT CrI₃.** **a** Zoomed-in view of a magnetization map measured on a selected sample area (400 nm × 400 nm) of the 0.15° tDT CrI₃ device, showing co-existing FM and AFM domains. **b** Schematic of NV spin relaxometry measurements to probe spin fluctuations of local FM and AFM states in a proximal sample. The blue and red arrows represent local magnetic moments forming spontaneous AFM and FM orders. Noncoherent magnetic noise arising from FM or AFM spin fluctuations at the NV ESR frequencies $f_{ESR}$ will drive NV spin transitions from the $m_s = 0$ to the $m_s = \pm 1$ state, resulting in enhanced NV relaxation rate. **c** Temperature dependence of NV spin relaxation rate $\Gamma$ measured when the NV center is positioned right above the FM and AFM domains formed in the 0.15° tDT CrI₃ sample. Control measurement results are also presented to characterize the intrinsic NV spin relaxation rate. **d** 1D NV spin relaxation rate $\Gamma$ measured along the white dashed lines across the co-existing FM and AFM domains in the 0.15° tDT CrI₃ device shown in Fig. 3a. The peak values of $\Gamma$ measured at 48 K and 58 K occur at the corresponding in-plane lateral positions of AFM and FM domains, respectively.

frequencies will induce NV spin transitions from the $m_s = 0$ to $m_s = \pm 1$ state, resulting in enhancement of the corresponding NV spin relaxation rates[27,32,33,36]. By measuring the spin-dependent NV photoluminescence, the occupation probabilities of NV spin states can be quantitatively obtained, allowing for extraction of NV spin relaxation rate that is proportional to the magnitude of the local fluctuating magnetic fields transverse to the NV axis (see Supplementary Information Note 6 for details)[27,32–34]. Figure 3c shows the measured temperature-dependent NV spin relaxation rate $\Gamma$ when the NV center is positioned right above the FM (AFM) domain formed in the 0.15° tDT CrI₃ sample (Fig. 3a). Due to the divergent magnetic susceptibility, the measured NV spin relaxation rate shows a clear enhancement across the second-order magnetic phase transition points of tDT CrI₃, from which the Curie and Néel temperatures of the co-existing FM-AFM states are measured to be ~58 K and ~48 K, respectively. We would like to highlight that similar experimental signatures are also observed in the 0.25° tDT CrI₃ sample as detailed in Supplementary Information Note 7. It is worth noting that twist-induced lattice reconstructions for small-twist angles typically happen at the boundary of different stacking areas within individual moiré supercells in tDT CrI₃[37], so the potential strain effect would play a marginal role in our measurements here.

At the monoclinic (AB') stacking sites, tDT CrI₃ features the same type AFM interlayer interaction across the six CrI₃ layers, thus it is not surprising that the formed local AFM moment manifests a similar $T_c$ with that of the atomically thin pristine CrI₃ crystals studied in previous work (see Supplementary Information Note 8 for details)[8,17,38]. In contrast, the FM order established in tDT CrI₃ is driven by rhombohedral

(AB) stacking featuring an enhanced, negative interlayer exchange coupling at the twisted interface[8,39]. Intuitively, such a stronger magnetic interaction will render the magnetic ordering more robust against thermal perturbations, resulting in an increased $T_c$ of the local FM state formed in tDT CrI₃. It is instructive to note that the $T_c$ (or $T_c$-equivalent) defined in the current manuscript for the FM order in tDT CrI₃, an "inhomogeneous" magnetic system along the thickness direction, has considered the overall magnetic contributions from all the six CrI₃ monolayers. Our one-dimensional (1D) scanning NV relaxometry measurements across the FM-AFM domains (Fig. 3d) further confirm this point. When $T = 48$ K, one can see that the measured 1D NV spin relaxation spectrum shows a peak value at the corresponding AFM domain site. As the temperature increases to 58 K, the observed peak of NV relaxation rate shifts to the FM domain side, demonstrating that the co-existing FM-AFM states show distinct magnetic phase transition temperatures. Note that such an experimental feature is absent in NV relaxation measurements performed at 38 K and 65 K, when the temperature is away from the Curie (Néel) points of FM (AFM) domains in 0.15° tDT CrI₃.

After showing the nanoscale stacking engineered $T_c$ of moiré magnetism, next, we present temperature-dependent magnetization maps to provide an alternative perspective to examine the second-order magnetic phase transitions in tDT CrI₃. Figure 4a, b presents reconstructed out-of-plane magnetization ($m_z$) maps of the 0.15° and 0.25° tDT CrI₃ devices measured at 38 K. It is evident that rhombohedral (AB) stacking induced FM state in tDT CrI₃ features uncompensated net magnetization. As the temperature increases, the FM moment in tDT CrI₃ gradually decreases and remains robust at 46 K

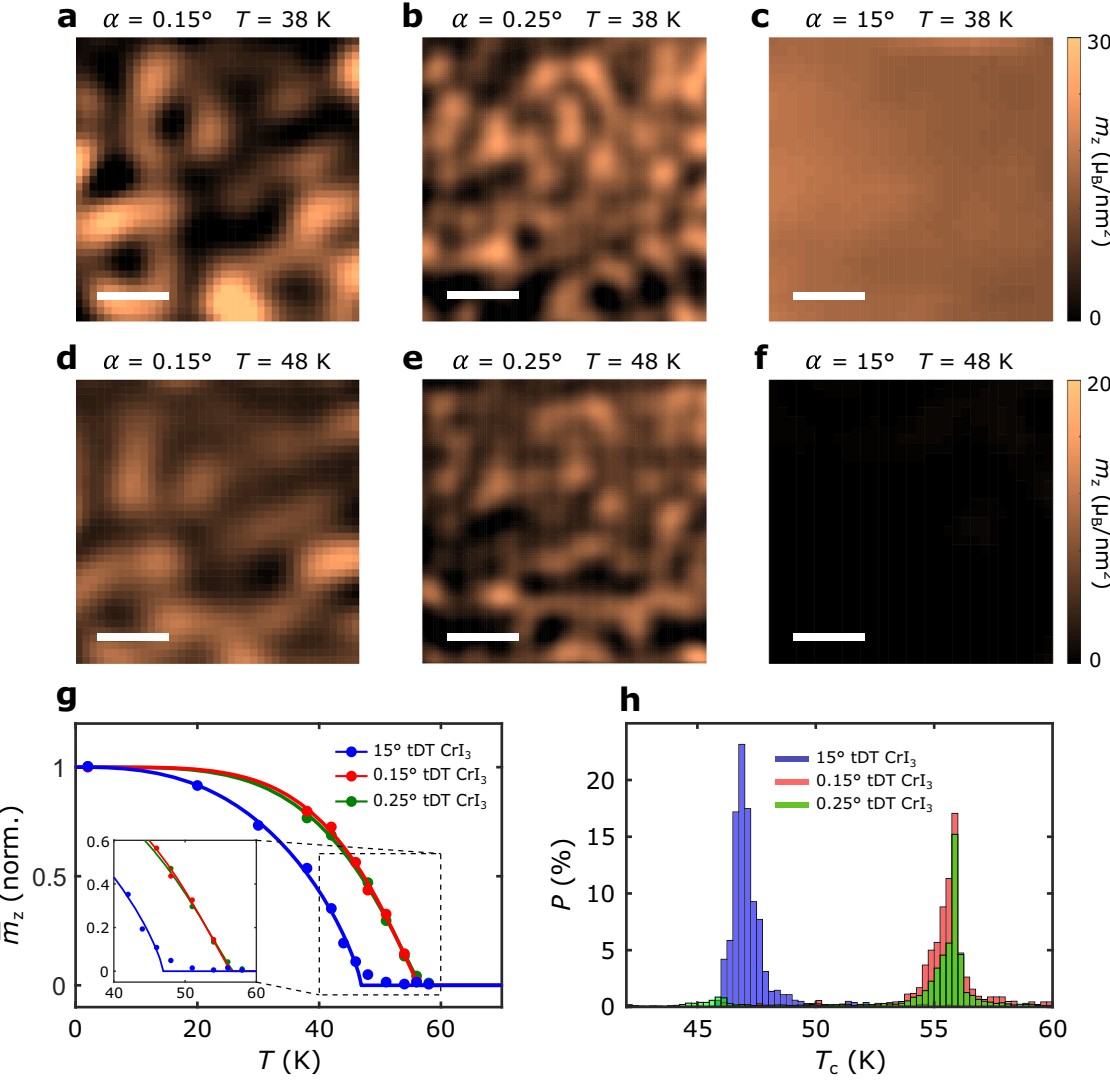

**Fig. 4 | Increased Curie temperatures of FM domains formed in small-twist-angle tDT CrI₃. a–c** Reconstructed magnetization maps of selected sample areas of 0.15° tDT CrI₃ (**a**), 0.25° tDT CrI₃ (**b**), and 15° tDT CrI₃ (**c**) device measured at 38 K. **d–f** Reconstructed magnetization maps of the same sample areas of the 0.15° tDT CrI₃ (**d**), 0.25° tDT CrI₃ (**e**), and 15° tDT CrI₃ (**f**) device measured at 48 K. Scale bar is 200 nm in **a–f. g** Temperature dependence of average out-of-plane magnetization $\bar{m}_z$ of FM domains formed in the selected sample areas of 0.15° tDT CrI₃, 0.25° tDT CrI₃, and 15° tDT CrI₃ device. Inset shows a zoomed-in view of the magnetic curves around transition temperatures. **h** Histograms of the obtained magnetic transition temperatures of individual FM domains formed in selected sample areas of 0.15° tDT CrI₃, 0.25° tDT CrI₃, and 15° tDT CrI₃ sample shown in **a–f**.

(Fig. 4d, e). Figure 4g plots the temperature dependence of the average out-of-plane magnetization $\bar{m}_z$ of FM domains formed in the selected sample areas of 0.15° and 0.25° tDT CrI₃ devices, from which the corresponding Curie point is obtained to be ~58 K, in agreement with our NV relaxometry results. Note that in the small-twist-angle regime, local moiré lattices relax to the monoclinic (AB') and rhombohedral (AB) stacking sites whose magnetic ground states follow their naturally preferred state. Under this condition, the interlayer exchange energy in tDT CrI₃ is dominated by the local stacking order, and the (small) twist angle plays a secondary role. Thus, the interlayer exchange energy of 0.15° and 0.25° tDT CrI₃ are basically the same, leading to the (almost) identical temperature-dependent magnetic phase transition behaviors. To further investigate stacking order-dependent magnetic response in tDT CrI₃, Fig. 4c presents a magnetization map of a 15° tDT CrI₃ device measured at 38 K. Notably, the co-existing FM-AFM phase disappears while a pure collinear FM ground state emerges in the large-twist-angle regime[1]. It is worth mentioning that the two CrI₃ trilayers are weakly ferromagnetically coupled at the twisted interface of 15° tDT CrI₃, resulting in a reduced Curie temperature (~48 K) in

comparison with that of the small-twist-angle tDT CrI₃. One can see that the 15° tDT CrI₃ sample enters the paramagnetic phase showing zero net FM moment at 48 K (Fig. 4f). To better illustrate this point, Fig. 4h presents the histograms of Curie temperatures of individual FM domains formed in 0.15°, 0.25°, and 15° tDT CrI₃ devices. Statistically, it is evident that FM domains in small-twist-angle tDT CrI₃ show clearly higher Curie temperatures than their counterparts in large-twist-angle tDT CrI₃ (see Supplementary Information Note 9 for details).

We now present a mean-field theoretical model of atomically layer-resolved magnetic order to explain the nanoscale stacking-dependent magnetic phase transitions observed in small-twist-angle tDT CrI₃. Our model consists of layer-uniform Ising spins representing out-of-plane magnetic moments of Cr atoms in tDT CrI₃ as shown in Fig. 5a. The intralayer magnetic exchange interaction $J_i$, stacking-dependent monoclinic (AB') type interlayer exchange interaction $J_M$, and rhombohedral (AB) type interlayer exchange interaction $J_R$ in tDT CrI₃ are obtained to be −1.32 meV, 0.086 meV, and −0.99 meV by comparing mean-field theory results with existing experimental transition temperatures and/or theoretically predicted values (See

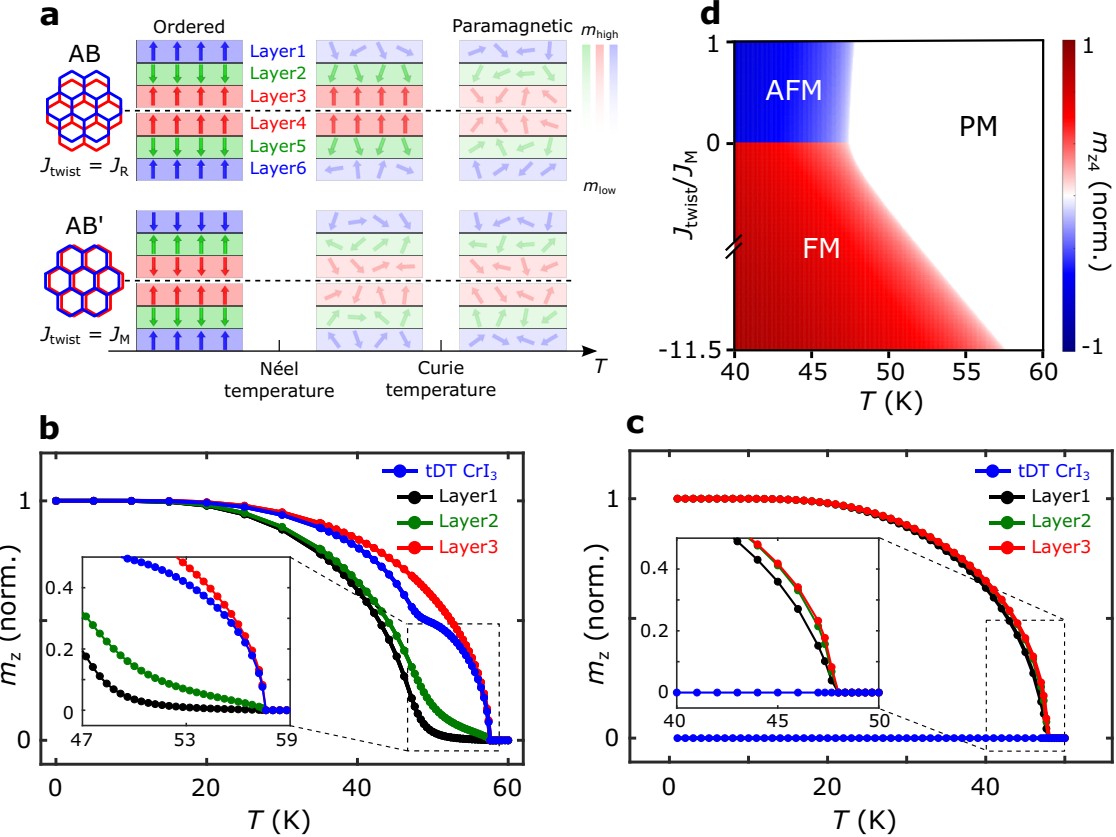

**Fig. 5 | Layer-resolved magnetic phases of small-twist-angle tDT CrI₃ with different stacking orders. a** Schematic of layer-resolved magnetic phases of rhombohedral (AB) and monoclinic (AB') stacked small-twist-angle tDT CrI₃ in ordered, intermediate, and paramagnetic state. The blue, red, and green arrows represent local magnetic moment carried by individual CrI₃ layers, and the fading background colors highlight the reduced magnetization with increasing temperature. Layers 1 to 6 are labeled from the top to bottom of tDT CrI₃ for reference. The black dashed lines highlight the twisted interface between the two CrI₃ trilayers. **b** Mean-field theory calculated temperature dependence of the normalized magnetization $m_z$ of layer 1 (black), layer 2 (green), and layer 3 (red) of small-twist-angle tDT CrI₃ with rhombohedral (AB) stacking sequence. Normalized total net magnetization $m_z$ curve of small-twist-angle tDT CrI₃ (blue) considering the overall contributions from the six CrI₃ layers is also presented. Inset shows a zoomed-in view of the features around the phase transition temperatures. **c** Calculated temperature-dependent variations of the normalized magnetization $m_z$ of layer 1 (black), layer 2 (green), and layer 3 (red) of small-twist-angle tDT CrI₃ with monoclinic (AB') stacking order. The total net magnetization curve of small-twist-angle tDT CrI₃ (blue) is also presented. Inset shows a zoomed-in view around the transition temperatures. **d** A constructed mean-field phase diagram of the normalized out-of-plane magnetization of CrI₃ layer 4 ($m_{z4}$) as a function of temperature $T$ and $J_{twist}/J_M$, highlighting the antiferromagnetic (AFM), ferromagnetic (FM), and paramagnetic (PM) states formed in small-twist-angle tDT CrI₃.

Method Section and Supplementary Information Note 10 for details)[40]. We compare the mean-field phases of tDT CrI₃ with uniform monoclinic stacking (bottom panel of Fig. 5a) and rhombohedral stacking at the twisted interface (top panel of Fig. 5a), approximating the situation for regions deep inside different stacking domains. The case of monoclinic stacking features the same exchange coupling between all neighboring layers and should, therefore, show a single Néel transition at the temperature scale set by $J_i$ and $J_M$. However, for the rhombohedral stacking case, the much stronger $J_R$ (in comparison with $J_M$) between the two middle CrI₃ layers (layer 3 and layer 4 shown in Fig. 5a) is expected to drive the system through an FM transition at a higher temperature. To corroborate the above physical picture, we have solved the mean-field equations of our model to get the temperature dependence of the normalized out-of-plane magnetization of individual CrI₃ layers as shown in Fig. 5b, c (See Supplementary Information Note 10 for details). In the rhombohedral (AB) stacking region (Fig. 5b), one can see that the middle CrI₃ layer (layer 3) indeed exhibits a higher $T_c \sim 58$ K in comparison with that of ~48 K for the monoclinic (AB') stacking case (Fig. 5c). Moreover, in the AB stacking case the mean-field magnetic moments of the two top CrI₃ monolayers (layer 1 and layer 2) undergo a smooth crossover as temperature decreases below $T_c$ before reaching the saturated value but not another phase

transition as dictated by the Lee-Yang theorem[41]. By considering the overall contributions from all the six CrI₃ layers, the effective Curie temperature ($T_c$-equivalent) of stacking induced FM order in small-twist-angle tDT CrI₃ is calculated to be 58 K, in agreement with our NV measurement results. In contrast, in the monoclinic (AB') stacking case (Fig. 5c), the top and middle CrI₃ monolayers (layer 1 and layer 3 in Fig. 5a) exhibit the same temperature dependence with a $T_c$ of ~48 K.

To further highlight the role of stacking engineering in affecting the local magnetic order and transition temperatures of small-twist-angle tDT CrI₃, Fig. 5d plots a mean-field phase diagram of the normalized out-of-plane magnetization of CrI₃ layer 4 ($m_{z4}$) as a function of temperature $T$ and $J_{twist}/J_M$ using our model. Here, $J_{twist}$ is a variable interlayer exchange interaction at the twisted interface of tDT CrI₃, and the sign of $m_{z4}$ is defined relative to the direction of the out-of-plane magnetization $m_{z3}$ of the CrI₃ layer 3. When $J_{twist}/J_M > 0$, the system orders antiferromagnetically layer-wise below the $T_c$ and $m_{z4}$ is oppositely aligned with $m_{z3}$ as shown in Fig. 5a. When $J_{twist}/J_M < 0$, the two twist-interfaced CrI₃ monolayers are ferromagnetically coupled and uncompensated net magnetic moment is formed in tDT CrI₃ below the magnetic critical temperature. Notably, as the magnitude of the FM-like coupling $J_{twist}$ increases, the calculated $T_c$ enhances from 48 K to 58 K when $J_{twist}/J_M$ reaches −11.5, corresponding to the

rhombohedral stacking case ($J_{\text{twist}} = J_R$). In the high-temperature regime, small-twist-angle tDT CrI$_3$ enters the paramagnetic phase where the long-range FM (AFM) order vanishes.

## Discussion

In summary, we have utilized scanning NV magnetometry techniques to investigate spatial and thermodynamic phase separation of moiré magnetism hosted by twisted CrI$_3$. By using NV spin relaxometry methods to probe local spin fluctuations, we explicitly show that the co-existing FM-AFM phases within individual moiré supercells manifest distinct second-order magnetic phase transition points. The measured Curie temperature of the rhombohedral (AB) stacking-driven FM state is significantly higher than the Néel temperature of monoclinic (AB′) stacking-driven AFM due to the spatially modulated exchange interaction at the twisted interface. In addition, we have directly visualized the stray field and magnetization maps across the magnetic phase transition points showing that twist engineering can effectively control the Curie temperature of local FM order in tDT CrI$_3$. Our results are well rationalized by a proposed mean-field theoretical model, which captures the layer-resolved $T_c$ in small-twist-angle tDT CrI$_3$. When extending to other combinations of atomically thin twisted odd number of CrI$_3$ layers, we expect that the observed stacking order-dependent magnetic phase transition with separate critical temperatures remains observable by the scanning NV quantum microscopy techniques. The presented results further highlight the opportunities provided by quantum spin sensors for investigating the local spin-related phenomena in moiré quantum magnets. The stacking and temperature-driven "intracell" magnetic phase separations observed in twisted 2D magnets may also find relevant applications in realizing local control of functional material properties through carefully engineered proximity effect, advancing the current state of the art of vdW spintronic devices[24,42].

## Methods

### Materials and device fabrications

CrI$_3$ crystals used in this study were grown by the chemical vapor transport method, as reported in previous literature[3]. The tDT CrI$_3$ devices are fabricated by the "tear-and-stack" method and encapsulated by hBN nanoflakes. The samples for SAED/TEM and scanning NV measurements were fabricated separately as described below, and different samples were used for SAED/TEM and NV studies. We first exfoliated the bulk crystals onto SiO$_2$/Si substrates to obtain trilayer CrI$_3$ and few-layer hBN. Next, the top hBN and one part of trilayer CrI$_3$ were picked up by a poly(bisphenol A carbonate) stamp by sequence. The other part of trilayer CrI$_3$ remained on the Si/SiO$_2$ substrate and was rotated by a well-controlled angle and then picked up. The two CrI$_3$ trilayer flakes were stacked with each other to form a twisted device and finally encapsulated by the bottom hBN flake. The layer number of atomically thin CrI$_3$ flakes was determined by thickness-dependent optical contrast and confirmed by magnetic circular dichroism measurements[5]. The entire 2D device fabrication processes were performed inside a nitrogen-filled glovebox with oxygen level below 0.1 ppm and water level below 0.5 ppm.

For scanning NV measurements, the final hBN/tDT CrI$_3$/hBN vdW stack was released onto quartz (SiO$_2$) substrates with prepatterned Au striplines. The quartz substrate was chosen due to its ideal insulating nature and dielectric properties for microwave performance. The Au stripline was utilized to deliver on-chip microwave currents to realize local control of the NV spin sensor. We have prepared multiple tDT CrI$_3$ samples to ensure the consistency of the presented NV results. In this paper, we reported scanning NV magnetometry studies of four CrI$_3$ devices including 0.15° tDT CrI$_3$, 0.25° tDT CrI$_3$, 15° tDT CrI$_3$, and a pristine trilayer CrI$_3$ control sample (presented in Supplementary Information). Note that samples with other layer thicknesses such as small-angle twisted double bilayer CrI$_3$ and twisted bilayer CrI$_3$, are not studied here due to the lack of net ferromagnetism or potential device quality issues. For SAED/TEM measurements, prepared hBN/tDT CrI$_3$/hBN stacks were transferred onto TEM grids with a 10-nm-thick SiN membrane. TEM images of tDT CrI$_3$ with different local twist angles are presented in Supplementary Information Note 2.

### Scanning NV magnetometry measurements

The presented quantum sensing measurements were performed using a scanning NV magnetometry system consisting of a home-built confocal and a custom-designed atomic force microscope (AFM) operating in a cryo-free cryostat (attocube Inc.). A commercially available diamond cantilever (Q-Zabre LLC) containing individually addressable NV centres was glued to a quartz tuning fork for force-feedback AFM operations, and a window on top of the cryostat provided optical access for NV measurements. A sample holder with coplanar waveguides was fixed onto a stack of piezo-based positioners and scanners to engage with the diamond cantilever and perform 2D scanning measurements. We applied continuous green laser and microwave signals to carry out NV optically detected magnetic resonance measurements[36]. NV spin states were addressed by measuring NV photoluminescence using an avalanche photodiode. Microwave signals delivered to the on-chip Au stripline were supplied by a Stanford Research Systems SG386 signal generator, and the external magnetic field applied in NV measurements was generated by a three-axis superconducting vector magnet.

### NV spin relaxometry measurements

Pulsed NV spin relaxometry measurements were performed using the scanning NV magnetometry system presented above. An external magnetic field of ~1200 G was applied along the NV spin axis in these measurements, and the corresponding NV ESR frequency for the $m_s = 0$ to $m_s = -1$ spin transition is ~0.5 GHz. Green-laser pulses used for NV initialization and readout were generated by an electrically driven 515-nm laser. The laser power entering the objective was ~0.5 mW. The trigger pulses to the optical modulator and photon counter were generated by a programmable pulse generator, and microwave signals were modulated by a switch (Minicircuits ZASWA-2-50DR+). The top panel of Supplementary Fig. 7b shows the details of the pulsed optical and microwave sequences for our NV spin relaxometry measurements. A 1.5-μs-long green laser pulse was first applied to initialize the NV spin to the $m_s = 0$ state. After a delay time $t$, we measured the occupation probabilities of the NV spin at the $m_s = 0$ and $m_s = -1$ state by applying a microwave $\pi$ pulse on the corresponding NV ESR frequencies and measuring the spin-dependent NV photoluminescence during the first ~600 ns of the green-laser readout pulse. By measuring the integrated photoluminescence intensity as a function of the delay time $t$ and fitting the data with a three-level model (See Supplementary Information Note 6 for details)[36], NV spin relaxation rates can be quantitatively measured.

### SAED and TEM measurements

We utilized Thermo Fisher Talos, operated at 200 kV and equipped with Gatan OneView camera, to perform the SAED and TEM measurements to characterize microscopic lattice structures of tDT CrI$_3$ devices encapsulated by hBN nanoflakes. The local mean twist angle of the sample was obtained by fitting 2D Gaussians to the Bragg peaks. Real-space structural image (Fig. 1c) was acquired by averaging DF-TEM images from three fifth-order Bragg peaks 120° apart. For tDT CrI$_3$, the moiré domains appear √3 smaller in the TEM image(s) in comparison to the expected domain size for the corresponding twist angle $\alpha$.

### Theoretical modeling

We used mean-field theory and minimal Ising models to investigate layer-resolved magnetic phases of small-twist-angle tDT CrI$_3$ with

different stacking orders. Each Ising spin, representing a Cr ion, has three intralayer nearest neighbors with exchange coupling $J_i$ and one nearest neighbor in an adjacent layer with interlayer exchange coupling $J_o$. The Hamiltonian of the system is given by:

$$H = \sum_{\langle jl \rangle a} J_i \sigma_{ja} \sigma_{la} + \sum_{j \langle ab \rangle} J_{o,ab} \sigma_{ja} \sigma_{jb} \tag{1}$$

where $j,l$ label sites within each layer and $a,b$ label layers, $\sigma_{ja} = \pm 1$, $J_i < 0$ (FM), and $J_{o,ab}$ can be negative (AB or rhombohedral stacking, denoted by $J_R$ below) or positive (AB' or monoclinic stacking, denoted by $J_M$ below) depending on the stacking order. Since we do not consider magnetic ordering beyond the FM order within each layer, $\sigma_{ja} \equiv m_{za}$ is independent of $j$. We fix $J_i, J_R$, and $J_M$ by requiring the mean-field $T_c$ of the Ising model to be consistent with the experimental (or theoretically predicted) values, from which the following exchange couplings: $J_i = -1.32$ meV, $J_R = -0.99$ meV, and $J_M = 0.086$ meV can be obtained. To mimic the local stacking order of small-twist-angle tDT CrI$_3$, we have considered different 6-layer models with AB' stacking between adjacent layers in the top 3 (labeled by layers 1, 2, 3) and bottom 3 (labeled by layers 4, 5, 6), but either AB or AB' stacking between layers 3 and 4. Temperature-dependent mean-field values of the layer-resolved Ising spins are solved numerically from the following nonlinear mean-field equation:

$$\langle m_{za} \rangle = \tanh \left[ -\beta \left( 3J_i \langle m_{za} \rangle + \sum_b J_{o,ab} \langle m_{zb} \rangle \right) \right] \tag{2}$$

## Data availability
All data supporting the findings of this study are available from the corresponding author upon reasonable request.

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

## Acknowledgements

The authors are grateful to Rainer Stöhr and Sreehari Jayaram for valuable discussions on cryogenic NV measurement techniques. This work was primarily supported by the U.S. Department of Energy (DOE), Office of Science, Basic Energy Sciences (BES), under award No. DE-SC0024870. Development of the cryogenic scanning NV microscopy was supported by the Air Force Office of Scientific Research (AFOSR) under Grant no. FA9550-20-1-0319 and its Young Investigator Program under Grant no. FA9550-21-1-0125. C.R.D. also acknowledges the support from the Office of Naval Research (ONR) under Grant no. N00014-23-1-2146. L.Z. acknowledges the support from the U.S. Department of Energy (DOE), Office of Science, Basic Energy Science (BES), under award No. DE-SC0024145 (for the moiré magnets sample fabrication) and AFOSR YIP Grant no. FA9550-21-1-0065 and Alfred. P. Sloan Foundation (for the instrument development of 2D vdW structure fabrication platform). A.S. and H.C. acknowledge the support from the National Science Foundation CAREER Grant no. DMR-1945023. R.H. acknowledges support from the Department of Energy, Basic Energy Sciences (DE-SC0024147). H.C.L. was supported by the National Key R&D Program of China (Grant no. 2018YFE0202600 and 2022YFA1403800), the Beijing Natural Science Foundation (Grant no. Z200005), the National Natural Science Foundation of China (Grants no. 12174443), and the Beijing National Laboratory for Condensed Matter Physics.

## Author contributions

S.L. performed the NV measurements and analyzed the data with N.M., M.H., H.L., and H.W.Z.S., and L.Z. prepared the $CrI_3$ devices. A.S. and H.C. performed the mean-field theoretical calculations. N.A., S.H.S., and R.H. performed the transmission electron microscopy characterizations. S.Y. and H.C.L. provided bulk $CrI_3$ crystals. C.R.D. supervised this project.

## Competing interests

The authors declare no competing interests.
