## [Peer Review File · Nature Communications]

Reviewers' Comments:

Reviewer #1:

Remarks to the Author:

The authors have done a good job of responding to all the referees' concerns. I strongly recommend publication in *Nature Communications*.

Reviewer #2:

Remarks to the Author:

Upon re-reviewing the revised manuscript by Li et al., the authors have sufficiently addressed my comments on the original submission and provided additional explanation to justify the conclusions of the manuscript. I recommend for publication in *Nat. Communications*.

Reviewer #3:

Remarks to the Author:

The authors gave satisfactory and unsatisfactory replies to different questions respectively. Below is my reading of the response:

Question #1: I am not convinced by the authors' reply. It seems that they, at current stage, can only produce "good quality" twisted trilayer CrI₃, which is the reason why they studied trilayer system alone. I understand even layers will not give net magnetization, but how about other odd layers. From the community, it appears other odd layer research exist and sometimes more appealing.

While being asked about whether the observation can be extended to other odd layers, the authors' answers are too brief without insights revealed. For example, when it is extended to other layer numbers, what should we expect in the layered dependent observation and why?

This question #1 is primarily to solicitate a broad understanding and applicability of this research. The authors are encouraged to answer in a more informative way and update the manuscript with these insights included, so even if no other odd layers are directly studied in this manuscript the audience can still learn from the valuable analysis.

Question #2: I don't feel the authors addressed the concern. What I was concerned about is that the "enhancement" appears meaningless in the context. If a material is engineered by external means such as doping and pressure, we can claim an enhancement when comparing a material property "before" and "after" an engineering. However, here are two local regions of two different exchange interactions (FM vs. AFM), how can we say "enhancement"? basically these two regions are two distinct material conditions such as interlayer stacking orders. When comparing two different materials, "enhancement" is not a meaningful term.

Question #3: I agree with the authors that the two samples (for SAED and scanning NV measurements) cannot be the same. Question is "can the authors produce a SAED sample with same twisting angle as the one for NV study?" They can be different samples, but both of the same or similar twisting angle).

Question #4: Here is the puzzling point. The authors are aware of the limitation of mean-field approaches in accuracy, but their mean-field calculations quantitatively reproduce the experimentally observed T_c/T_n difference by simple parameters. This indicates that the simple

parameterization did too much than what they can do. So, can we still trust that mean-field approach here can give a clear physical picture by taking in the wrong parameter (by saying "wrong", I assume this parameter should not be able to replicate T_c/T_n difference but it surprisingly can now in authors' calculation. This suggests the parameters contain lots of extra information that this parameter itself should not contain, loosing or distorting the original meaning of the parameter)

Question #5: It is unfair to quote the two references to highlight the authors' 10K difference is dramatic, for the two references were to engineer a material but here the two regions are fundamentally corresponding "two material parameters and conditions". This goes back to the previous comment that the FM and AFM regions are two distinct regions meaning two different materials. Should not use the difference of these two regions to claim "enhancement" or to compare the author's 10K with the T_c change in the two references.

Question #6: This question was very fundamental, how can one define a T_c or T_n of an inhomogeneous system, here including two single-phase trilayers with complex inter-trilayer interactions? The authors did not directly answer this question. I suggest they tune down the wording or redefine something like a quasi- T_c or T_c -equivalent. These suggested may not be good wording as well. But the authors should be careful not to use a single T_c to define an inhomogeneous system. This question and the authors' reply do not affect the result part of this work but can strengthen the scientific rigor.

Question #7: short answer is sufficient. The authors have addressed well.

The final question: the final questions asked about the true understanding of this work, instead of just reporting an observation and some tentative explanation of what was seen. These questions asked about what are the twisting angle dependence (the authors answered no dependence) and what fundamental parameters are critical for the reported observation of the twisted systems. It appears, from the authors reply, that there is not such well derived knowledge. This seems a weakness of a strong scientific article and will limit its broad impact, for the audience cannot extract the concrete knowledge to design similar experiments and plan the possible outcomes.

Response to reviews – NCOMMS-24-09427A

We thank the Reviewers 1 and 2 for their recommendation of our work. In this revision, we have addressed the Reviewer 3's additional comments, taken his/her constructive suggestions, and revised our manuscript accordingly. We hope that Reviewer 3 finds our answers thorough, reasonable, and clear. Below, we will provide a detailed point-by-point response. The reviewers' comments are displayed in *italic* format. Our response is in black (Times New Roman) and changes in the main text and Supplementary Information are highlighted in **red**.

Reviewer #1:

The authors have done a good job of responding to all the referees' concerns. I strongly recommend publication in Nature Communications.

Response: We thank the Reviewer for the recommendation of our manuscript.

Reviewer #2:

Upon re-reviewing the revised manuscript by Li et al., the authors have sufficiently addressed my comments on the original submission and provided additional explanation to justify the conclusions of the manuscript. I recommend for publication in Nat. Communications.

Response: We are glad that the reviewer is satisfied with our response and revision. We thank the Reviewer again for his/her recommendation of our manuscript.

Reviewer #3:

The authors gave satisfactory and unsatisfactory replies to different questions respectively. Below is my reading of the response:

Response: We thank the reviewer for devoting time and efforts to evaluate our manuscript again. Reviewer's technical questions will be addressed in detail below.

Question #1: I am not convinced by the authors' reply. It seems that they, at current stage, can only produce "good quality" twisted trilayer CrI₃, which is the reason why they studied trilayer system alone. I understand even layers will not give net magnetization, but how about other odd layers. From the community, it appears other odd layer research exist and sometimes more appealing.

Response: The stacking order dependent magnetic phase transitions with separate critical temperatures within moiré supercell(s) is largely an interfacial effect, thus, a thinner sample should naturally result in a more prominent experimental effect. As mentioned in our first-round response, twisted bilayer (1L + 1L) CrI₃ devices may have additional issues brought by the reduced crystalline and magnetic quality of monolayer and twisted bilayer CrI₃, which could cause complications to data interpretation of our scanning NV imaging measurements. In fact, it is not a sample/device issue that only we have. This point was also experimentally reported by other leading research group(s) including Dr. Xiaodong Xu and his colleagues in their seminar work *Science* **374**, 1140 (2021).

We are aware of the appealing research on other twisted odd CrI₃ layer systems in this community, however, after balancing the sample thickness and quality consideration, we are convinced that twisted double trilayer (3L + 3L) CrI₃ provides a reasonably suitable material platform for the purpose of the current work.

While being asked about whether the observation can be extended to other odd layers, the authors' answers are too brief without insights revealed. For example, when it is extended to other layer numbers, what should we expect in the layered dependent observation and why?

This question #1 is primarily to solicitate a broad understanding and applicability of this research. The authors are encouraged to answer in a more informative way and update the manuscript with these insights included, so even if no other odd layers are directly studied in this manuscript the audience can still learn from the valuable analysis.

Response: We thank the reviewer for this constructive point. When extending to other combinations of twisted odd layer number of CrI₃, we expect that the observed stacking order dependent magnetic phase transition with separate critical temperatures remains observable by the scanning NV quantum microscopy techniques. It is instructive to note that a thinner sample will give a more pronounced experimental effect as discussed above. Following the reviewer's suggestion, we have included these insights in the last paragraph on page 6 as follows:

“..... When extending to other combinations of atomically thin twisted odd number of CrI₃ layers, we expect that the observed stacking order dependent magnetic phase transition with separate critical temperatures remains observable by the scanning NV quantum microscopy techniques.”

Question #2: I don't feel the authors addressed the concern. What I was concerned about is that the “enhancement” appears meaningless in the context. If a material is engineered by external means such as doping and pressure, we can claim an enhancement when comparing a material property “before” and “after” an engineering. However, here are two local regions of two different exchange interactions (FM vs. AFM), how can we say “enhancement”? basically these two regions are two distinct material conditions such as interlayer stacking orders. When comparing two different materials, “enhancement” is not a meaningful term.

Response: We understand the concern brought by the reviewer on this potential language issue. In this revision, we have further polished the relevant expressions to downplay the expression of “enhancement” of T_c in our manuscript. Please find the detailed changes highlighted in red color in the revised main text.

Question #3: I agree with the authors that the two samples (for SAED and scanning NV measurements) cannot be the same. Question is “can the authors produce a SAED sample with same twisting angle as the one for NV study?” They can be different samples, but both of the same or similar twisting angle).

Response: We believe that we have addressed this specific question in our previous response and revision. The twist angle α of a twisted double trilayer (tDT) (3L + 3L) CrI₃ sample for scanning NV measurements shown in Fig. 2f (main text) is $\sim 0.25^\circ$. On a separate tDT CrI₃ sample with a very similar local twist angle (α) of $\sim 0.28^\circ$, we also have reported TEM characterizations as shown in Supplementary Fig. 2f. We hope that this satisfactorily answers the reviewer's concern.

Question #4: Here is the puzzling point. The authors are aware of the limitation of mean-field approaches in accuracy, but their mean-field calculations quantitatively reproduce the experimentally observed T_c/T_n difference by simple parameters. This indicates that the simple parameterization did too much than what they can do. So, can we still trust that mean-field approach here can give a clear physical picture by taking in the wrong parameter (by saying "wrong", I assume this parameter should not be able to replicate T_c/T_n difference but it surprisingly can now in authors' calculation. This suggests the parameters contain lots of extra information that this parameter itself should not contain, losing or distorting the original meaning of the parameter)

Response: We thank the reviewer's comments. Here, we would like to kindly point the reviewer to our response to Reviewer #2's Question #2 in the last round of report. Basically, the exchange coupling parameters of the spin model are chosen so that they are not only able to reproduce the experimental transition temperatures for the tDT CrI₃ systems, but also are not far off when compared to existing literature results. The typical overestimation of transition temperatures from mean-field theory translates to, roughly speaking, underestimation of the exchange coupling parameters. However, such quantitative differences do not affect the qualitative picture provided by the mean-field theory, which is the stacking-dependent phase transition and the temperature dependence of the layer-resolved ordered spin below the transition temperature. We believe the revised Supplementary Note 10 now contains a candid discussion on the limitations and benefits of our mean-field approach.

Question #5: It is unfair to quote the two references to highlight the authors' 10K difference is dramatic, for the two references were to engineer a material but here the two regions are fundamentally corresponding "two material parameters and conditions". This goes back to the previous comment that the FM and AFM regions are two distinct regions meaning two different materials. Should not use the difference of these two regions to claim "enhancement" or to compare the author's 10K with the T_c change in the two references.

Response: We thank the reviewer again for this detailed point. As stated in our response above, we have downplayed the expression of "enhancement" of T_c in the revised manuscript.

Question #6: This question was very fundamental, how can one define a T_c or T_n of an inhomogeneous system, here including two single-phase trilayers with complex inter-trilayer interactions? The authors did not directly answer this question. I suggest they tune down the wording or redefine something like a quasi- T_c or T_c -equivalent. These suggested may not be good wording as well. But the authors should be careful not to use a single T_c to define an inhomogeneous system. This question and the authors' reply do not affect the result part of this work but can strengthen the scientific rigor.

Response: We understand the reviewer’s concern on using a single T_c to define an “inhomogeneous” material system. Following the reviewer’s suggestion, we have refined the relevant expressions to tune down the wording to strengthen the scientific rigor of this manuscript.

“.....It is instructive to note that the T_c (or T_c -equivalent) defined in the current manuscript for the FM order in tDT CrI₃, an “inhomogeneous” magnetic system along the thickness direction, has considered the overall magnetic contributions from all the six CrI₃ monolayers.”

“.....By considering the overall contributions from all the six CrI₃ layers, the effective Curie temperature (T_c -equivalent) of stacking induced FM order in small-twist-angle tDT CrI₃ is calculated to be 58 K, in agreement with our NV measurement results.”

Question #7: short answer is sufficient. The authors have addressed well.

The final question: the final questions asked about the true understanding of this work, instead of just reporting an observation and some tentative explanation of what was seen. These questions asked about what are the twisting angle dependence (the authors answered no dependence) and what fundamental parameters are critical for the reported observation of the twisted systems. It appears, from the authors reply, that there is not such well derived knowledge. This seems a weakness of a strong scientific article and will limit its broad impact, for the audience cannot extract the concrete knowledge to design similar experiments and plan the possible outcomes.

Response: We thank the reviewer again for the detailed and constructive evaluation of our manuscript. Here, it is worth pointing out that although we do not expect significant twist angle dependent variations of major physical parameters in the small twist angle regime, it remains an open and highly interesting scientific question when it comes to the intermediate and large twist angle regime in tDT CrI₃. We hope that our study will stimulate more research interests and efforts in this fast-growing research topic, advancing the current understanding of emergent electromagnetic behaviors in moiré quantum matters. We are convinced that the immediate and broad impact of the current work on condensed matter physics and quantum sensing community is clear.

Reviewers' Comments:

Reviewer #3:

Remarks to the Author:

I am satisfied with the answers to all my questions except the question #4 on "mean-field approach". I don't think the wrong calculation of T_c by mean-field approach in other previous work is because of the wrong adoption of exchange interaction value. Basically, mean field approaches cannot capture 1) all the detailed exchange interactions (various exchange interactions between 1st, 2nd, 3rd-nearest neighbors) and 2) excitation details (various magnon modes and magnon interaction with other particles such as electrons and phonons). So, even a mean-field approach with the so-called "correct exchange interaction" still cannot predict the correct T_c or T_n . From this end, fundamentally mean-field approach is a coarse method but not a scientifically rigorous approach, nevertheless with significant physics overlooked. I am sure the authors cannot justify the rigor of this approach. Thus I recommend to move this mean-field calculation and analysis to SI just as some "tentative" explanation. With this part downplayed (which is absolutely necessary), I recommend its publication. We cannot overclaim this coarse method too much, since it is well known that one parameter (i.e., exchange interaction) cannot address the complicated T_c calculation that involves very complex excitations at non-zero temperatures. As said, with this part properly downplayed, I am fine with its publication.